# Engineering Polypropylene–Calcium Sulfate (Anhydrite II) Composites: The Key Role of Zinc Ionomers via Reactive Extrusion

**DOI:** 10.3390/polym15040799

**Published:** 2023-02-05

**Authors:** Marius Murariu, Yoann Paint, Oltea Murariu, Fouad Laoutid, Philippe Dubois

**Affiliations:** 1Laboratory of Polymeric and Composite Materials, Materia Nova Materials R&D Center & UMONS Innovation Center, 3 Avenue Copernic, 7000 Mons, Belgium; 2Laboratory of Polymeric and Composite Materials, Center of Innovation and Research in Materials and Polymers (CIRMAP), University of Mons (UMONS), Place du Parc 20, 7000 Mons, Belgium

**Keywords:** polypropylene, PP, mineral-filled composites, calcium sulfate, gypsum, anhydrite II, melt mixing, thermal and mechanical properties, reactive extrusion, REx, heat deflection temperature (HDT), injection molding, automotive, engineering applications

## Abstract

Polypropylene (PP) is one of the most versatile polymers widely used in packaging, textiles, automotive, and electrical applications. Melt blending of PP with micro- and/or nano-fillers is a common approach for obtaining specific end-use characteristics and major enhancements of properties. The study aims to develop high-performance composites by filling PP with CaSO_4_ β-anhydrite II (AII) issued from natural gypsum. The effects of the addition of up to 40 wt.% AII into PP matrix have been deeply evaluated in terms of morphology, mechanical and thermal properties. The PP–AII composites (without any modifier) as produced with internal mixers showed enhanced thermal stability and stiffness. At high filler loadings (40% AII), there was a significant decrease in tensile strength and impact resistance; therefore, custom formulations with special reactive modifiers/compatibilizers (PP functionalized/grafted with maleic anhydride (PP-g-MA) and zinc diacrylate (ZnDA)) were developed. The study revealed that the addition of only 2% ZnDA (able to induce ionomeric character) leads to PP–AII composites characterized by improved kinetics of crystallization, remarkable thermal stability, and enhanced mechanical properties, i.e., high tensile strength, rigidity, and even rise in impact resistance. The formation of Zn ionomers and dynamic ionic crosslinks, finer dispersion of AII microparticles, and better compatibility within the polyolefinic matrix allow us to explain the recorded increase in properties. Interestingly, the PP–AII composites also exhibited significant improvements in the elastic behavior under dynamic mechanical stress and of the heat deflection temperature (HDT), thus paving the way for engineering applications. Larger experimental trials have been conducted to produce the most promising composite materials by reactive extrusion (REx) on twin-screw extruders, while evaluating their performances through various methods of analysis and processing.

## 1. Introduction

Polypropylene (PP) is one of the most used thermoplastics available for almost all end-use markets, ranging from packaging and textiles to automotive and engineering components [1,2,3,4]. Nowadays, the global markets of PP are showing outstanding growth, driven by technological progress and an increase in the number of applications. This is due to the low density of PP, its good chemical resistance, and the related thermal, mechanical, and electrical properties of the PP-based products, as well as their excellent processability and recyclability [5,6]. Moreover, in many engineering applications, metals have been extensively replaced by PP-based materials to achieve weight reductions, low energy/fuel consumption, and cost savings [7,8]. Still, regarding the current development status of PP, it is important to mention the announcement of the production of bio-based PP made from eco-friendly building blocks derived from natural resources [6]. Taking as one key example the use of PP in the automotive sector, a non-exhaustive list of applications includes various interior and exterior components: pillars, consoles, armrests, air cleaner bodies, carpet fibers, dashboard components, flexible bumpers with high impact resistance, engine fans, wheel covers, instrumental panels and door trims, radiator headers, heater baffles and housings, tanks, electrical parts, battery boxes, etc. [2,9,10].

The addition to the PP matrix (homopolymer or PP-based copolymers) of reinforcing fibers (e.g., short or long glass fibers, various natural fibers) [4,11,12,13], nano- and micro-fillers [14], impact modifiers [13,15], other dispersed phases and special additives, such as flame retardants [16,17], is known as an important route to achieving the enhancement of PP properties required for specific end-use applications, and in some cases, improved processing and cost savings. The mineral-filled (MF) PP compounds can offer enhancements over the unfilled PP, in properties such as stiffness, thermal and dimensional stability, heat deflection at higher temperatures, better processing, and cost-effectiveness [18]. Moreover, they can be designed for a wide range of applications: engineering parts, automotive components, packaging, production of cups and containers, transport pallets, and others.

PP has been melt blended with clays [19,20,21], CaCO_3_ [8,22,23,24,25], talc [26,27,28,29], mica [30,31], kaolin [28,32,33,34], and other mineral fillers [35,36] typically used in the polymer composites industry. Talc and CaCO_3_ are among the most preferred fillers used to produce PP compounds. At low loadings, talc can act as a nucleating agent, reducing PP spherulite size and shortening the processing time, while at higher loadings (up to 40 wt.%) it behaves as a reinforcing filler, significantly increasing the tensile modulus and stiffness of PP [37,38,39]. On the other hand, there is a dramatic decrease in the strain at break and impact resistance [39], especially at high talc percentage, with the latter parameter being of great importance for many PP applications. To improve the proprieties of MF PP composites, different experimental strategies have been applied: using fillers with various surface treatments [40,41,42]; compatibilizing agents [43,44,45] such as PP functionalized/grafted with maleic anhydride (PP-g-MA) [46,47]; using elastomers [26]; special modifiers/additives [48]; following the addition and synergies between micro- and/or nano-fillers [49,50,51,52], and so on.

Nonetheless, few laboratory-scale studies have considered the use of synthetic CaSO_4_ (CS) particles to reinforce PP [53,54,55,56]. Unfortunately, the information on the use of stable CS for the reinforcement/filling of PP is scant by comparison to other fillers (talc, CaCO_3_, kaolin, etc.), while the nature of CS (available in different forms), or the necessity of specific thermal treatments, has been much less reported on. In addition, we believe there is a misunderstanding related to the rapid water absorption or high moisture sensitivity that are specific to CS hemihydrate and to “soluble” β-anhydrite III (AIII), both derived from CaSO_4_·2H_2_O (gypsum), as obtained by thermal treatments above 100 °C and at ca. 200 °C, respectively [57]. On the contrary, compared to these not stable forms, calcination of gypsum at higher temperatures (e.g., at 500–700 °C in an industrial process), allows us to obtain highly stable β-anhydrite II (AII), also known as “insoluble” anhydrite [57,58,59].

Related to this study, it is worth mentioning that nowadays the producers of natural gypsum look for new markets by offering CS derivatives, such as stable AII, for new applications with higher added value, e.g., for the industry of polymer composites, paints, and coatings. In this respect, CS anhydrite AII, which is characterized by high thermal stability and whiteness, is unfortunately not well known by the polymer composite industry. Consequently, further prospects are required to demonstrate its beneficial value with different polymer matrices and by considering the needs of a large range of applications. It is also important to note that AII has been used with promising results in the manufacturing of polylactic acid (PLA) composites even though this aliphatic (bio)polyester is known for its high thermal and hydrolytic sensitivity [57,60]. Accordingly, the use of natural gypsum, and particularly of its derivatives (i.e., AII), which are available in massive quantities, could be an interesting option to produce new PP-based composites and an answer to the current industrial requests.

Starting from the state of the art, the main goal of this paper is to present the experimental pathways followed to produce, characterize, and design the properties of new MF PP–AII composites containing up to 40 wt.% AII (thus issued from natural gypsum). Regarding the choice of PP as the matrix, it is well known that PP homopolymers are considered for engineering applications because they are characterized by higher crystallinity and improved chemical resistance compared to PP-based copolymers. Nevertheless, they are stiffer and have better strength at high temperatures, but unfortunately, their impact resistance is somewhat limited [61], so this parameter must be carefully considered. Interestingly, it has been reported elsewhere that the filling of PP with rigid fillers (CaCO_3_, talc, others), results in increased rigidity/stiffness, but in some cases, better impact properties can also be achieved [62,63]. Therefore, it is of great interest to have information about the properties of PP–AII composites and to identify their most relevant features, linked to the requests of end-use applications (i.e., from packaging to automotive/engineering sectors).

In this study, first, to obtain small quantities of materials, PP was filled by melt compounding with 20–40 wt.% AII (β-CaSO_4_) using internal mixers. Then, the PP–AII composites were characterized to obtain primary information on their properties. These composites (without any modifier) have shown quite good filler distribution within the PP matrix, enhanced thermal stability, and stiffness. To compensate for the decrease in tensile and impact strength, after the high filling of PP with AII (e.g., at 40 wt.% filler), it was necessary to propose custom formulations. For compatibilization and to obtain enhancements of proprieties, the PP/AII compositions have been modified with selected reactive modifiers, such as PP functionalized/grafted with maleic anhydride (PP-g-MA), and an ionomeric monomeric additive, zinc diacrylate. In fact, fillers such as talc, calcium carbonate, and AII are polar in nature; therefore, for better interfacial adhesion (PP–filler), the modification of PP can be accomplished by attaching polar groups onto the PP backbone, e.g., via the addition of PP-g-MA [64,65], or using metallic ionomers (i.e., zinc diacrylate to lead to Zn ionomer) [66,67], experimental methods often considered in prior studies.

The effects of filling PP with AII and reactive modification were deeply evaluated in terms of morphology, mechanical and thermal properties, to allow the selection of key compositions. The formation of clusters/ionic crosslinks of Zn ionomer and the favorable interactions with the filler allowed an impressive increase in the properties of PP–AII composites. Moreover, in the frame of current prospects, the most interesting composites in terms of properties were extrapolated on twin-screw extruders (TSE) to produce larger quantities of materials by reactive extrusion (REx) and to make them available for processing by injection molding (IM). The PP–AII composites produced by REx have been characterized using several methods of analysis to highlight their performances: high tensile and flexural strength, stiffness, enhanced storage modulus under dynamic mechanical solicitations, and superior HDT. By considering the overall features of these new PP–AII composites, it is expected that they will show great promise as potential novel products designed especially for technical/engineering applications requiring rigidity, heat resistance, dimensional stability, and improved processing.

## 2. Materials and Methods

### 2.1. Materials

PP homopolymer (Moplen HP400R, produced by LyondellBasell), is a PP grade for IM, suitable for food contact applications, characterized by high fluidity and good stiffness. The characteristics of interest according to the technical sheet of the product are: density: 0.9 g/cm^3^, melt flow rate (MFR): 25 g/10 min (230 °C, 2.16 kg), tensile strength (yield): 32 MPa, tensile strain at break: >50%, and Charpy impact strength (notched): 2 kJ/m^2^.

CaSO_4_ β-anhydrite II (AII) mineral filler, delivered as “TOROWHITE Ti-ExR04”, was kindly supplied by Toro Gips S.L. (Spain). According to the information provided by the supplier, AII is obtained from high-purity natural gypsum. It is characterized by high whiteness/lightness (L *), AII being an alternative of choice as a white pigment (TiO_2_) extender. Color measurements performed in the CIELab mode (illuminate D65, 10°) with a SpectroDens Premium (TECHKON GmbH, Königstein, Germany) proved the high lightness of the AII sample, i.e., L * of about 98.0.

PP functionalized/grafted with maleic anhydride (PP-g-MA), modified with 8–10% maleic anhydride (MA), was supplied by Sigma-Aldrich. The molecular weights (M_w_ and M_n_) of PP are, respectively, 9100 and 3900 Da. This product, evaluated only in the primary stage of the study, is abbreviated hereinafter as PPMA1.

Polybond 3200 (supplier Chemtura, abbreviated as PPMA2) is a PP homopolymer modified with maleic anhydride (MA) with the following characteristic features: MFR (190 °C, 2.16 Kg): 115 g/10 min, melting point: 157 °C, and MA content: 0.8–1.2%. PPMA2 was used only as an alternative for producing reference samples by REx.

Zinc diacrylate (produced as Dymalink^®^ 9200 by Total Cray Valley) is an ionomeric monomeric additive recommended for the modification of polyolefins that allows resin producers and compounders to impart ionomeric character to their materials [68,69]. In unreinforced, filled, and thermoplastic polyolefin (TPO) systems, this reactive modifier can lead to an increase in melt strength, enhanced mechanical properties, and high-temperature performances. Dymalink^®^ 9200 was made available as an off-white powder (100% active) with the following characteristic features: molecular weight: 207, specific gravity: 1.68, and functionality: 2. The product will be abbreviated as ZnDA. For more clarity, the chemical structures of the two modifiers tested in this study are shown in Figure 1.

### 2.2. Production of PP–AII Composites

#### 2.2.1. Melt Compounding with Internal Mixers

Before melt compounding, all materials were dried overnight (PP and all modifiers at 70 °C, and the AII filler was dried at 100 °C). PP–AII composites were obtained by melt compounding at 220 °C, using a Brabender bench scale internal mixer (W50EHT, Plastograph EC, Brabender GmbH &. Co. KG, Duisburg, Germany) equipped with “came” blades. Conditions of processing: feeding at 30 rpm for 4 min, followed by melt mixing at 100 rpm for 10 min. For the sake of clarity, the codes and compositions of filled PP composites produced with internal mixers are shown in Table 1. Neat PP processed under similar conditions has been used as one key reference. The so-obtained PP composites were processed by compression molding (CM) at 210 °C, using an Agila PE20 hydraulic press to obtain plates (100 mm × 100 mm~3 mm thickness). More specifically, the material was first maintained at low pressure for 180 sec (3 degassing cycles), followed by a high-pressure cycle at 150 bars for 120 sec. Then, the cooling was realized under pressure (50 bars) for 300 sec using tap water (temperature slightly > 10 °C). The plates produced by CM were used to obtain specimens for mechanical characterizations. Throughout this contribution, all percentages are given as weight percent (wt.%).

#### 2.2.2. Reactive Extrusion (REx) Using Twin-Screw Extruders (TSE)

In the subsequent experimental step, larger quantities of the most promising formulations/composites were produced by REx, using a Leistritz twin-screw extruder (TSE) as the equipment (ZSE 18 HP-40, L (length)/D = 40, diameter (D) of screws = 18 mm). Before melt compounding, PP and the AII filler were dried overnight at 70 °C and 100 °C, respectively, using drying ovens with recirculating hot air. The experimental setup used to produce PP–AII–ZnDA composites in larger quantities is shown in Figure 1.

The filler (AII) was previously mixed with ZnDA into a Zeppelin Reimelt Henschel FML4 mixer (Zeppelin Systems GmbH—Henschel Mixing Technology, Kassel, Germany). Basically, to produce PP filled with 20% and 40% AII, the filler was premixed with ZnDA powder (AII/ZnDA weight ratio of 20/2 or 40/2, respectively) for 15 min at 1500 rpm. Two separate gravimetric feeders were used for the dosing of PP (granules) and AII/ZnDA powders. It is noteworthy that to evaluate PPMA2, two controlled feeders were used, a first one to feed the blends of PP and PPMA2, after a previous premixing, and a second one for AII. The parameters of REx/compounding were as follows: (a) temperatures on the heating zones of TSE: Z1 = 180 °C, Z2 = 190 °C, Z3-Z5 = 215 °C, Z6 = 200 °C, and Z7 = 195 °C; (b) die of extrusion = 180 °C; (c) speed of the screws = 170 rpm; (d) throughput = 3 kg/h.

The samples produced by REx as granules have been dried overnight at the temperature of 70 °C. Specimens for tensile, flexural, impact, and heat deflection temperature (HDT) characterizations were produced with a Babyplast 6/10 P injection molding (IM) machine, using adapted processing temperatures (e.g., Z1 = 200 °C; Z2 = 205 °C; and Z3 (die) = 200 °C; temperature of the mold = 30–40 °C).

### 2.3. Methods of Characterization

(a) Thermogravimetric analyses (TGA) were performed using a TGA Q50 (TA Instruments, New Castle, DE, USA) by heating the samples under air or nitrogen from room temperature (RT) up to maximum of 800 °C (platinum pans, heating ramp of 20 °C/min, 60 cm^3^/min gas flow). In TGA, the initial degradation temperature (i.e., T_5%_, °C), was considered as the temperature at which the weight loss is 5 wt.%, whereas the temperature of the maximum degradation (T_d_) was defined as the temperature at which the samples present the maximal mass loss rate (data obtained from D-TG curves).

(b) Differential scanning calorimetry (DSC) measurements were conducted using a DSC Q200 from TA Instruments (New Castle, DE, USA) under nitrogen flow. The traditional DSC procedure was as follows: first, heating scan (10 °C/min) from 0 °C to 220 °C, isotherm at this temperature for 2 min, and then cooling by 10 °C/min to −50 °C, and, finally, a second heating scan from −50 to 220 °C at 10 °C/min. The first scan was used to erase the prior thermal history of the polymer samples. The events of interest linked to the crystallization of PP during the DSC cooling scan, i.e., the peaks of crystallization temperature (T_c_) and the enthalpies of crystallization (ΔH_c_), were quantified using TA Instruments Universal Analysis 2000 software—Version 3.9A (TA Instruments—Waters LLC, New Castle, DE, USA). It is noteworthy that all data are normalized to the amounts of PP from the samples. The thermal parameters were also evaluated in the second DSC heating scan and abbreviated as follows: melting peak temperature (T_m_), melting enthalpy (ΔH_m_), and final DC (χ). The DC (degree of crystallinity) was determined using the following general equation:$$\chi \;=\frac{\Delta \mathrm{Hm}}{\Delta H\begin{array}{c} 0\\ m \end{array}\times W_{\mathrm{PP}}}\times 100\;\left(\%\right)$$
where ΔH_m_ is the enthalpy of melting, W is the weight fraction of PP in composites, and $\Delta H_{m}^{0}$ is the melting enthalpy of 100% crystalline PP considered 207 J/g [70].

(c) Mechanical testing: Tensile tests were performed with a Lloyd LR 10K bench machine (Lloyd Instruments Ltd., Bognor Regis, West Sussex, UK), according to ASTM D638-02a norm on specimens-type V, at a typical speed of 10 mm/min (specimens of 3.0–3.2 mm thickness). The flexural properties were determined using a three-point bending test and the NEXYGEN program (Lloyd Instruments Ltd.). The measurements were performed on rectangular specimens (80 mm × 10 mm × 4 mm) at a testing speed of 10 mm/min, by adapting the Lloyd LR 10K tensile bench with bending grips (span = 60 mm), in accordance with ISO 178. For the characterization of Izod impact resistance, rectangular specimens (63 mm × 12 mm × 3.2 mm), a Ray-Ran 2500 pendulum impact tester, and a Ray-Ran 1900 notching apparatus (Ray-Ran Test Equipment Ltd., Warwickshire, UK) were used, according to ASTM D256 norm (method A, 3.46 m/s impact speed, 0.668 kg hammer). All mechanical tests were performed on specimens previously conditioned for at least 48 h at 20 ± 2 °C, under a relative humidity of 50 ± 3%, and the values were averaged over at least five measurements.

(d) Scanning electron microscopy (SEM) analyses on previously cryo-fractured PP samples at a liquid nitrogen temperature were performed using an FE-SEM SU-8020 Hitachi instrument equipped with triple detectors (Hitachi, Tokyo, Japan), at various accelerated voltages and magnitudes. For better insight and easy interpretation, the SEM analyses were performed with detectors for both secondary and backscattered electron imaging (SE and BSE, respectively). Reported microphotographs represented typical morphologies as observed at, at least, three distinct locations. SEM analyses (SE mode) were also performed on the surfaces of selected specimens fractured by tensile or impact testing to have more information about their behavior under mechanical solicitations. SEM combined with energy-dispersive X-ray spectroscopy (EDX) was also used to obtain information about the elemental composition of selected samples and/or to reveal the elemental atomic distribution within the PP matrix.

(e) Rheological measurements: The melt volume/flow rate (MVR/MFR) was determined only on samples produced by REX (as granules) following the procedure described in ASTM D1238, using a Davenport 10 Melt Flow Indexer (AMETEK Lloyd Instruments Ltd., West Sussex, UK) at a temperature of 190 °C, with a 2.16 kg load.

(f) Heat deflection temperature (HDT) measurements were performed according to ISO 75 norm, using an HDT/Vicat 3–300 Allround A1 (ZwickRoell GmbH & Co, Ulm, Germany) equipment. The specimens for testing (80 mm × 10 mm × 4 mm), were produced by IM. All measurements were evaluated under a load of 0.45 MPa and at a heating rate of 120 °C/h using at least three specimens.

(g) DMA (dynamic mechanical analysis) was performed on rectangular specimens obtained by IM (63 mm × 12 mm × 3 mm) using a DMA 2980 apparatus (TA Instruments, New Castle, DE, USA), in dual cantilever bending mode. The dynamic storage and loss moduli (E′ and E″, respectively) were determined at a frequency of 1 Hz and amplitude of 20 µm, in the range of temperature from −75 °C to −150 °C at a heating rate of 3 °C/min.

(h) Color measurements (L *, a *, b *) were performed in the CIELab mode (illuminate D65, 10°) with a SpectroDens Premium (TECHKON GmbH, Königstein, Germany).

## 3. Results and Discussion

### 3.1. Preliminary Considerations

It is important to note that CS is available in several forms: dihydrate—CaSO_4_·2H_2_O (commonly known as gypsum), hemihydrate—CaSO_4_ 0.5H_2_O (Plaster of Paris, stucco, or bassanite), and different types of anhydrites [57,59,71]. The CS phases obtained by progressive dehydration and calcination of gypsum are in the following order [59]: dihydrate → hemihydrate → anhydrite III → anhydrite II → anhydrite I (the last one obtained at temperatures higher than 1180 °C). Dehydration of gypsum above 100 °C at low pressure (vacuum) or under air (at atmospheric pressure) favors the formation of β-CS hemihydrate. Increasing the temperature to around 200 °C enables the production of β-anhydrite III—which is not stable, whereas as mentioned in the introductory part, the calcination at higher temperature (e.g., at 500–700 °C) allows obtaining “insoluble” β-anhydrite II (AII), characterized by extremely slow rates of rehydration [57,59].

Figure 2a,b shows selected SEM pictures to illustrate the morphology of AII microparticles used in this study. The granulometry of AII was characterized by dynamic light scattering, using a Mastersizer 3000 laser particle size analyzer (Malvern Panalytical Ltd., Malvern, UK), the microparticles having a Dv50 of 3.6 µm and a Dv90 of 12.9 µm.

The particulate filler is characterized by a low aspect ratio, whereas a shared morphology, i.e., particles with irregular shape and a fibrillar/flaky aspect, due to the cleavage of CS layers [59] during the production/grinding process, are typically evidenced by SEM analysis. It is worth mentioning that the rough surfaces can increase the number of anchorage points with the polymer matrix, thus offering good filler–resin mechanical interlocking, which can influence the interfacial adhesion and mechanical properties of composites. To allow the use of CS in the production of polymer composites, we will retell that it is of prime importance to dry (dehydrate) the CS dihydrate or hemihydrate prior melt compounding, or it is required to use stable anhydrite forms, such as AII. Indeed, it was reported elsewhere that β-AII (made from natural or synthetic gypsum), is much better suited for melt blending with a polymer sensitive to the degradation by hydrolysis, e.g., the case of polyamides or polyesters, such as polylactic acid (PLA) [57,58,72].

The high purity of the AII sample was proven by SEM-EDX analysis, which confirms the presence of Ca, S, and O as the main atomic elements, and only of few traces of carbon (EDX spectra shown in Figure 3a). Furthermore, the excellent thermal stability of the filler up to 700 °C is confirmed by TGA (Figure 3b). Hence, by its addition to the PP matrix, it is expected to have beneficial effects on the thermal properties of PP composites.

For a first evaluation, the PP–AII compositions from Table 1 (see the experimental part), have been produced in small quantities using internal mixers, whereas, in the next experimental step, the most representative composites have been extrapolated on twin-screw extruders (TSE) and deeply characterized for the evidence of key properties.

### 3.2. PP–AII Composites Produced within Internal Mixers

After the addition of mineral fillers to PP, it is expected that a great part of properties will be improved (e.g., stiffness, thermal stability, aesthetic), while, on the other hand, other properties might decrease to some extent (tensile strength, impact resistance). To maximize the benefits of MF PP composites, it is necessary to understand and combine the relationship between the properties of the matrix and characteristics of dispersed phases (fillers, additives, etc.), to improve their compatibility and interactions, to control the stabilizing or degradation effects, and the influence of the manufacturing process on the final product characteristic features, and so forth.

#### 3.2.1. Morphology of Composites

For better evidence of the filler distribution state through the PP matrix, SEM imaging was performed on all composites (20% and 40% filler) using backscattered electrons (BSE) to obtain a higher phase contrast. Traditionally, it is expected to obtain better individual particles dispersion at lower filler loading than at high filling [58,72], and this assumption is confirmed by the SEM analyses of PP composites filled with 20% AII (SEM micrographs shown in the Appendix A). Figure 4a–d show representative SEM-BSE images at different magnifications of cryo-fractured surfaces of PP and of highly filled PP–AII composites containing 40% filler, a loading at which the filler could show some poorer distribution/dispersion.

The BSE technique has a high sensitivity to the differences in atomic number, giving information about the distribution of CS (i.e., the presence of Ca atoms is evidenced by brighter zones). Well-distributed particles, with various geometries and quite broad size distribution, are evidenced at the surface of cryo-fractured PP composites, even though the amount of filler is high (i.e., 40%). It is worth mentioning that large aggregates are not observed, whereas such a quality of distributive mixing within the hydrophobic PP matrix is obtained without any previous surface treatment of the hydrophilic filler. Still, from the SEM pictures at higher magnification, it is evident that the AII particles are characterized by a low aspect ratio and irregular shape of micrometric size. On the other hand, the morphology of PP–40AII–ZnDA appears slightly different compared to that of other composites, even though all blends are performed using internal mixers. In fact, the SEM images of this sample suggest better wetting/surrounding by the PP matrix, of well-distributed and even well-dispersed AII microparticles. The improved morphology can be ascribed to a more distinct compatibilizing effect (PP–AII), following the reactive modification.

#### 3.2.2. Thermal Properties

Obviously, AII is a mineral filler characterized by high thermal stability. In the absence of undesirable catalytic degradation effects, its incorporation into different polymers is expected to result in composites characterized by similar, or even better thermal stability, with respect to the neat polymer matrix. Indeed, as shown from the comparative TGA traces shown in Figure 5a,b, the filling with, 20% and 40% AII, respectively, clearly result in a delay in the thermal degradation of PP, well determined by the amounts of filler. Furthermore, a more pronounced stabilization effect is achieved for PP–AII–ZnDA composites, i.e., after the co-addition of AII and ZnDA (chemically grafted as Zn ionomer). In fact, the TGA data (Table 2) confirm the boosting of both thermal parameters, i.e., T_5%_ and T_d_.

The significant rise in the onset of thermal degradation (T_5%_) and of maximum decomposition temperature (T_d_, from max. D-TG), is assigned to the presence of filler, with a mention for the PP–AII composites modified with ZnDA, which exhibits the best thermal properties. Indeed, the PP–AII–ZnDA samples show the shift of T_d_ above 60 °C compared to the neat PP processed under similar conditions. On the other hand, there is no evidence of an additional effect linked to the use of PP-g-MA (PPMA1), because the composites PP–AII and PP–AII–PPMA1 display similar thermal behavior. Still, the increase in thermal stability of PP following the co-addition of AII and ZnDA (T_5%_ of PP–20AII–ZnDA and PP–40AII–ZnDA is increased by 25 °C and 49 °C, respectively, compared to PP), is considered as a key property in the perspective of the application of such materials. These enhancements can be ascribed at least in part to a better filler dispersion (as was revealed by SEM images), but the influence of other factors is not excluded, e.g., a stabilizing effect linked to the formation of Zn ionomer.

Figure 6a,b show the comparative DSC traces of the neat PP and those of PP–AII composites obtained during DSC cooling from the molten state (a), and following the second DSC heating by 10 °C/min (b). From the analysis of DSC results obtained during non-isothermal crystallization (Figure 6a, DSC data shown in Table 3), it emerges that the nucleation/kinetics of crystallization of PP is improved in presence of AII (T_c_ of PP–AII composites is at about 120 °C, higher than that of neat PP, i.e., 116 °C). On the other hand, when using PPMA1 as a modifier, the crystallization of PP is slightly delayed (e.g., PP–20AII–PPMA1 has a T_c_ of 115 °C). In contrast, the composites reactively modified with 2% ZnDA (i.e., PP–AII–ZnDA) exhibit a T_c_ at 126 °C that is significantly higher compared to that of neat PP. A similar behavior was reported elsewhere for PP modified with Surlyn ionomers, by Ma et al. [73]. The crystallization rate was accelerated by the ionic aggregates/clusters of ionomers which were reported to initiate the heterogeneous nucleation of PP. As a result, T_c_ increased, and the crystallization process was faster for the PP/ionomer systems with respect to the neat PP.

Still, regarding the second DSC heating (Figure 6b, data gathered in Table 3), a slight increase in T_m_ is noted for the composites that have shown higher T_c_ (i.e., PP–AII and PP–AII–ZnDA), and that are finally characterized by somewhat higher DC. Moreover, because PP homopolymer inherently has high rates of crystallization, the values of DC for all samples are high and remarkably close to each other, i.e., in the range of 45–48%, which makes it difficult to identify the glass transition temperatures (T_g_) under the specific conditions of DSC analysis. However, the crystallization mechanisms of these composites require more understanding (e.g., insight into the crystal nucleation/structure and its growth, α-nucleation, and occurrence with β-nucleation), via different techniques of investigation (DSC, polarized light microscopy, X-ray diffraction, etc.); therefore, we will refrain here from more comments. Further studies are under consideration.

#### 3.2.3. Mechanical Properties

In general, it is expected that the reinforcement of PP with mineral fillers will lead to improved mechanical properties, such as increased stiffness (i.e., higher tensile and flexural modulus, enhanced flexural strength), whereas, on the other hand, the reduction of tensile strength, of strain at break and impact resistance, is often reported [39].

The evolution of the tensile and impact properties of PP and PP–AII composites (reactively modified or not) is shown in Figure 7a–d. By analyzing the effects of AII addition into PP (without any modifier), is noted that the ultimate (maximum) tensile strength (σ_t_) of the PP (37 MPa) is gradually diminished to 29 MPa and 25 MPa, adding 20% and 40% filler, respectively (Figure 7a). On the other hand, when using PP-g-MA as a compatibilizer, only limited improvements are seen, especially for the highly filled composites (i.e., PP–40AII–PPMA1), characterized by a σ_t_ of about 29 MPa. On the contrary, the most interesting properties are revealed by PP–AII–ZnDA composites (σ_t_ of 32–34 MPa). However, the stress–strain curves (Figure 7b) give more insight into the mechanical behavior during tensile testing of PP composites. Accordingly, a stronger reinforcing effect by filling is obtained for PP–AII composites using ZnDA than PPMA1, which was found to be less effective as a compatibilizer, under the specific experimental conditions used in this study. However, it is not excluded that the low molecular weight of PPMA1 can explain the decrease in mechanical parameters; therefore, the testing of other PP-g-MA products was also considered (more information shown elsewhere). Although the elongation at the break (**ε_b_**) of specimens obtained by CM decreased drastically, especially at high filling, from 11% (neat PP) to 3–6% for composites containing 40% AII. Nevertheless, currently there is still a need for a better understanding of the mechanisms that can explain the enhancements of properties of PP–AII composites after the reactive modification with ZnDA (i.e., interactions between Zn ionomer and filler, formation of clusters or ionic crosslinking networks, liable for better thermo-mechanical properties). Moreover, the interfacial adhesion (polymer filler), is considered to be one of the main factors affecting the strength of the composites [62]. In connection with the interfacial properties (PP–AII) in the different composites, it is worth mentioning that for better insight, supplementary SEM analyses have been performed on the surfaces of fractured specimens by mechanical testing. The SEM micrographs suggest a stronger interfacial adhesion between PP and filler in the case of PP–AII–ZnDA composites (numerous regions of contact and better wetting of AII by the PP matrix, the particles are more deeply lodged within the polymeric matrix, etc.), rather than the debonding at the interface, which is more specifically seen for the other composites (SEM images shown as Appendix A, and representative examples in Section 3.3.1.).

Regarding the evolution of rigidity/Young’s modulus (Figure 7c), it is reasonable to consider that this parameter is determined by the amounts of filler, e.g., a progressive increase of 19% and 42% compared to the neat PP is obtained for PP–AII composites containing, respectively, 20% and 40% AII. Here, the effect of ZnDA is less evident, maybe because this parameter (Young’s modulus) is determined in the limit of elasticity at low strain/deformation.

In terms of impact properties (Figure 7d), unfortunately, PP homopolymers are relatively “brittle”, requiring low crack propagation energy for breaking (impact resistance of 2.3 kJ/m^2^, on specimens performed by CM). In several cases, it has been reported that the addition of rigid fillers can have positive effects on the impact resistance of polymers, and PP is included [62]. This improvement was also stated for PLA composites filled with 20% AII, which are characterized by higher impact resistance than the neat PLAs [57]. Interestingly, the PP–20AII sample shows a little bit higher impact resistance than the neat PP matrix. At higher filler amounts (40% AII), a dramatic decrease in this parameter is observed for the unmodified composite (PP–40AII). On the other hand, notable enhancements are seen after the reactive modification with ZnDA (impact resistance in the range of 3–3.5 kJ/m^2^, much higher than that of neat PP). Accordingly, ZnDA leads to the most noteworthy enhancement of impact/toughness, whereas the modification with PPMA1 was found, again, to be less effective. The increase in impact resistance of PP modified with ionomers is elsewhere ascribed to the decrease in the spherulite size of PP, to a specific/unique structure and morphology [73]. The ionomers allow the formation of crosslink points and networks of molecular chains, which could absorb the impact energy and prevent crack initiation.

#### 3.2.4. Key Considerations and Findings

The overall experimental results reveal the key role of the reactive modification with metallic acrylate monomer (ZnDA) for the enhancement of thermal and mechanical properties of PP–AII composites. On the contrary, PP-g-MA (i.e., PPMA1) was found to be less effective in improving the compatibility/interfacial properties between the PP matrix and filler, and the final characteristics of the composites. To the best of our knowledge, the PP–AII–ZnDA composites/formulations are entirely new; therefore some more comprehension will be required to explain their performances using additional methods and techniques of characterization, not concerned by this paper (i.e., Fourier transform infrared (FTIR) and X-ray photoelectron spectroscopy (XPS), rheometric analyses, and the use of predictive mathematical models for composite mechanical properties). It is also noteworthy to mention that the SEM-EDX analyses performed on PP–AII–ZnDA composites give evidence for the fine distribution/dispersion of elemental Zn within the PP matrix (an illustrative example is shown in Figure 8).

It is important to note that the use of ZnDA together with peroxides to cure and improve the rubber’s properties, has been applied in industry for decades [68,74,75]. On the other hand, the addition of ionomeric additives, such as ZnDA, to conventional PP is reported to improve its melt strength, and also other properties (hardness, impact strength, HDT, chemical resistance, gas barrier, etc.) [69,76,77,78]. High melt-strength PP-based homopolymers and copolymers are required for IM, thermoforming, and foaming applications. Moreover, ionic crosslinking (e.g., with Zn acrylates) is of interest in the recycling process to strengthen, toughen, and compatibilize polymer blends [79].

Zn acrylates are reported to form ionomeric crosslinks with the polyolefin (PP) chains. These crosslinks are “thermo-reversible”, allowing the material to be processed using traditional equipment (IM machines, extruders, etc.) while providing significant enhancements of properties at temperatures below 150 °C, that are of interest in the case of engineering materials. Regarding the mechanism of the reaction, it is considered that the addition of zinc acrylate monomers during melt mixing/extrusion at temperatures above 210 °C allows acrylate double bonds to graft on PP chains. Then, the polar zinc salts (Zn ionomer) could associate to form ionomeric crosslinks during cooling [80]. Hence, in the frame of this study, it was assumed that the conditions of reactive melt blending using an internal mixer (i.e., melt blending for 10 min, at 220 °C) represent a good experimental choice to allow the grafting of ZnDA on PP chains and to lead to high density crosslinked structures through ionic bonds. Moreover, the reaction between PP and ZnDA is feasible even in the absence of peroxides [80], which are commonly used as initiators for the grafting on PP chains, or for the long-chain branching PP [81,82]. In fact, PP is very sensitive to thermo-mechanical/oxidative degradation at temperature; therefore, the easy generation of radicals and formation of degradation products by β-scission [83,84] during melt mixing can be assumed, even in the absence of peroxides.

Figure 2a suggests, based on the information from the state of the art, the formation of zinc salt clusters following the grafting of ZnDA onto PP chains (formation of Zn ionomer), ionic domains deemed as thermo-reversible “crosslinks” at temperatures ≤ 190 °C [80,85]. Furthermore, to explain the enhancements of properties obtained by the co-addition of AII and ZnDA, Figure 2b proposes an alternative structure, based on the hypothesis that the Zn ionomer and AII filler can also interact via ionic bounds (e.g., ion pairs, metal–ion coordination) [75,86], leading to the formation of ionic clusters/crosslinks, physical entanglements/aggregates, and supramolecular networks. Indeed, from the category of multivalent ions, calcium is considered available for ionic crosslinking and chelation with carboxylate groups (−COO**^−^**) [87,88].

The ionic interactions and/or the creation of new ionic bonds/interactions, i.e., between Zn^2+^ and Ca^2+^ as cations, with SO_4_^2**−**^ and −COO**^−^** anions, respectively, could be (hypothetically) responsible for the good compatibility/interfacial properties between PP and filler, and finally, can explain the thermo-mechanical performances of these new custom composites, i.e., PP–AII–ZnDA. It is also important to be precise that for additional information, an unfilled PP composition modified with 2% ZnDA was prepared and characterized under similar conditions. Compared to the neat PP, the mechanical properties remain almost comparable, except for the impact resistance, which has increased from 2.3 kJ/m^2^ (neat PP) to about 3 kJ/m^2^ for the reactive modified PP (results shown in Appendix A).

For potential utilization in engineering applications, it emerges that the PP–AII–ZnDA composites are characterized by the most interesting performances. Obviously, the formation of ionic crosslinks via the addition of ZnDA (formation of Zn ionomer) and the strong interactions between components play a key role to strengthen (σ_t_, 32–34 MPa) and toughen the PP–AII blends. The dramatic decrease in impact resistance by high filling (only 1.4 kJ/m^2^ for PP–40AII composites), is clearly corrected to higher values using ZnDA (>3 kJ/m^2^), which is significantly better than that of pristine PP (2.3 kJ/m^2^). Accordingly, these tailored PP–AII–ZnDA composites have been proposed for upscaling by REx using TSE (details hereinafter).

### 3.3. Current Prospects: Production of PP–AII Composites by REx

For the supplementary confirmation of the results obtained with laboratory mixers (Section 3.2), and in the perspective of upscaling on pilot plants, it was decided to use twin-screw extruders (TSE) to produce higher quantities of PP–AII–ZnDA composites, and to process them by IM. Nevertheless, regarding the presumed differences between the reactive melt mixing with internal mixers and TSE, it is believed that the temperatures of melt compounding, the shear and residence time, and the conditions of processing used to obtain final products (e.g., IM vs. CM) are among the factors that are requiring increased attention.

#### 3.3.1. Characterization of PP–AII Composites Produced by REx

For a simplified reading and more comprehension, the discussion hereinafter is focused on two key compositions produced in larger quantities, to allow their processing by IM, i.e., PP–20AII–ZnDA (TSE) and PP–40AII–ZnDA (TSE). These PP composites are filled with 20% and 40% AII, respectively, and both are modified with 2% ZnDA (details in the experimental section). For comparative reasons, the unfilled PP (processed on TSE under similar conditions) is used as a reference. As was mentioned in the experimental part, the realization of PP–AII composites modified with PPMA2 (i.e., PP containing 0.8–1.2% grafted MA) was also considered, but unfortunately, the mechanical properties (i.e., the tensile strength and impact resistance) were less promising than using ZnDA (results shown in Appendix A).

Hereinafter are summarized the key results obtained following the production and analyses of these novel composites (PP–AII–ZnDA), mostly to confirm the good reproducibility of results obtained using different reactive melt-mixing procedures:

(a) Morphology of composites: Figure 9 shows representative SEM images of the cryo-fractured surfaces of PP–AII–ZnDA composites produced by REx. From the micrographs obtained at low and higher magnification, it emerges that following the melt blending/REx at shorter residence time and higher shear with TSE, once more, AII has a good distribution within the PP matrix, since there is not any evidence for the presence of agglomerates of microparticles, even at high filing.

(b) Thermal properties: TGAs attest again that, by the progressive increase in filler amounts up to 40 wt.%, the thermal stability of PP is remarkably improved (Figure 10a), e.g., the onset of thermal degradation (T_5%_) is increased by more than 50 °C for PP–40AII–ZnDA (TSE) composites with respect to the neat polymer. These enhancements are ascribed to the effects of the filler and its outstanding distribution/dispersion, and, to a stabilizing effect linked to the presence of Zn ionomer, following the grafting by REx of ZnDA onto PP chains. Moreover, the DSC analyses reconfirm the previous findings obtained with laboratory internal mixers. From the analysis of DSC traces obtained during non-isothermal crystallization (DSC cooling, Figure 10b), it is seen that the samples containing AII and Zn ionomer show higher T_c_ compared to the neat PP (T_c_ of 129 °C and only 116 °C, respectively). By considering overall DSC results, it can be assumed that both components (AII and ZnDA) act as effective nucleating agents for PP, allowing improved kinetics of crystallization, a propriety of interest for better processing. On the other hand, regarding the second DSC heating, there is only a slightly noticeable increase in T_m_ for PP–AII–ZnDA (TSE) composites, from 163 °C to 165 °C.

(c) Mechanical properties: Because the materials have been produced in larger quantities by REx, the specimens required for mechanical characterizations were obtained by IM processing. Table 4 shows the comparison of the main properties of PP and PP–AII–ZnDA (TSE) composites, including their mechanical properties. Obviously, the PP–AII composites modified by REx with ZnDA show excellent tensile strength (σ_t_ of 34–35 MPa), whereas their rigidity/Young’s modulus is progressively increased in correlation with the amounts of AII, e.g., by about 70%, by filling PP with 40% AII. These results are in good agreement with those obtained in the selection phase of key compositions, following the experimental tests with internal mixers.

In the context of engineering applications, it is important to note that the high-filled composites (40% AII) exhibit a higher flexural strength (65% greater than that of neat PP), while the flexural modulus is almost twice as high for composites compared to the neat PP (2550 MPa vs. 1160 MPa), mechanical properties which may be relevant for the realization of automotive and mechanical components. Then again, the PP–AII–ZnDA (TSE) composites show better impact resistance than the neat PP (2.7–2.8 kJ/m^2^, compared to 1.9 kJ/m^2^, respectively). IM specimens are, however, slightly less impact-resistant than the CM specimens, highlighting the significance of processing conditions. Additionally, it was found that **ε_b_** of PP was higher after IM than following the processing by CM, the difference was assigned to the better orientation of PP chains by IM.

As was mentioned elsewhere, the good mechanical properties of PP–AII composites can be ascribed not only to the effects of crosslinking, due to the formation of clusters of Zn ionomer, but also to the existence of good compatibility between the PP matrix and filler. Additionally, it is not excluded that the pendant ionic groups in the ionomer might increase the polarity of PP [66,67,89], and this plays a key role in improving the interactions with AII. As in the case of other ionomers, Zn ionomer can act as a compatibilizer, allowing a finer filler dispersion and better adhesion between the PP and AII microparticles [66,67,90].

Nevertheless, for better evidence of the interfacial properties (PP–filler), additional SEM investigations have been performed on the fractured surfaces of composites, obtained after impact and tensile testing (selected images shown in Figure 11a,b, respectively). First, it is important to note that during tensile testing the composites are stretched at low speed (10 mm/min) until they break. In impact tests, a sudden shock is given to the material at a high energy/deformation rate (respectively, 3.9 J and 3.46 m/s), and this leads to the fracture of PP specimens. By observing the SEM images of the fractured surfaces following the impact solicitation (Figure 11a), it is seen that the filler has quite good adhesion to the PP matrix, as the pull-out of particles from the matrix and/or signs of debonding at the interface polymer filler are not observed. Moreover, the distribution/dispersion of AII microparticles and their good wetting/surrounding by the polymer matrix is noticeable (Appendix A).

On the other hand, regarding the specimens fractured at lower speed by tensile testing (Figure 11b), together with the evidence of ductile/plastic deformation specific to PP, the presence of numerous zones of contact at the interface between the polymeric matrix and AII microparticles, as fibrils or elongated/deformed plastic regions, is reasonably ascribed to the existence of good adhesion/interfacial properties, which finally reflect the level of mechanical performances.

(d) Rheological properties: Following the evolution of MVR (data shown in Table 4), it emerges that the melt viscosity of PP composites is progressively rising (NB: MVR is decreased) in correlation with the AII amounts. Nevertheless, under the specific conditions of testing (190 °C), an additional effect assigned to the presence of Zn ionomer, linked to the formation of ionic crosslinks that are able to increase the melt strength of PP, is not excluded. However, more comprehensive rheological studies can highlight other features of interest, to allow the optimal processing of these composites using different techniques: injection molding (IM), which requires high fluidity and increased IM temperatures; extrusion, which needs increased viscosity and good melt strength.

#### 3.3.2. Other Properties of Interest for Engineering Applications

Figure 12 shows selected images to illustrate the aesthetic aspect/whiteness of samples as granules and as specimens produced by IM, together with the results of color analysis. The color measurements (CIELab) confirm the increase in the lightness (L *) of the composites in direct correlation with the amounts of filler (N.B. the AII used in this study has the L * = 98; therefore, it is also used as the TiO_2_ extender).

From the perspective of engineering applications, the high rigidity of modified PP–AII composites was also proved under dynamic mechanical solicitations by DMA (Figure 13a,b). The tests performed in the range of temperatures −75 °C–+150 °C, highlight the possibility to use these composites at higher mechanical stress and temperature, than the neat PP. For illustration purposes only, Table 5 shows comparative values of storage modulus (E’, Figure 13a) at different temperatures.

On the other hand, the loss modulus (E“, Figure 13b) displays a main peak at about 16 °C for all samples (assigned to a β-relaxation), whereas the rising of amounts of filler to 40% leads to the progressive increasing of E” in all range of temperatures, and this is mainly ascribed to the contribution of the mechanical loss generated in the interfacial regions [91]. Furthermore, the DMA results agree with those of HDT testing (Figure 14), which evidence properties that are relevant for engineering applications, since the HDT of neat PP (about 70 °C) is significantly increased to 100–110 °C in the case of filled PP composites.

Before concluding, it is important to point out that the results of REx are highly dependent on the type of equipment and the conditions of processing. Experimental fine-tuning is highly recommended to find the optimal conditions/compositions. Nevertheless, the distinct conditions of production (use of internal mixer or TSE, differences in temperatures and residence time) and processing (CM or IM), could lead to some changes in the final properties of composites. Nonetheless, the overall experimental results obtained in this study are in good agreement. However, in the frame of further prospects, it would be important to reconfirm the performances of these novel composites and to obtain additional validations regarding their behavior under different conditions/temperatures of utilization, or on the crystallization mechanisms, via alternative techniques of investigation (polarized light microscopy, X-ray diffraction, etc.). Nevertheless, starting from the requirements of the application, it is considered that some more specific characterizations will be more necessary (e.g., evaluation of aging at elevated temperature or under UV conditions, tests of permeability, abrasion, scratch resistance, and so on).

## 4. Conclusions

The study addresses current requests regarding the utilization of natural products (i.e., gypsum derivatives) to produce new high-performance mineral-filled PP composites designed for engineering applications. First, using melt blending internal mixers, the effect of filling PP with up to 40% CaSO_4_ β-anhydrite II (AII) has been evaluated in terms of morphology, mechanical and thermal properties. The PP–AII composites (without any modifier) showed quite good filler distribution within the PP matrix, enhanced thermal stability, and stiffness. However, especially at high filling (40% AII), the impact resistance and tensile strength were drastically reduced. Therefore, for compatibilization and enhanced properties, the addition of reactive modifiers (PP-g-MA and ZnDA) was considered. Finer filler dispersion and surprising thermal and mechanical properties were obtained following the co-addition of AII and ZnDA into PP, rather than using PP-g-MA.

PP–AII–ZnDA composites show remarkable performances: improved kinetics of crystallization, high thermal stability, and excellent mechanical properties (tensile strength, rigidity, improved/surprising impact resistance). It was presumed that the formation of clusters/ionic crosslinks of Zn ionomer, the favorable interactions with the filler, and the good interfacial properties allowed us to compatibilize, strengthen, and toughen the composites. Moreover, experimental trials by REx have been performed to produce larger quantities of materials for processing by injection molding and to reconfirm their properties. Due to high thermal stability and notable mechanical performances, the PP–AII–ZnDA composites proved attractive for use in technical applications: tensile strength of about 35 MPa, improved stiffness/Young’s modulus, impact resistance higher than that of PP, advanced rigidity under dynamic solicitations (DMA) in the range of temperatures from −75 °C to + 150 °C, and the increasing of HDT from 70 °C (neat PP) to 110 °C (for composites). Based on the performances of these new composites, it is anticipated that due to their specific features, they will be highly sought after as potential new candidates for technical/engineering applications, requiring rigidity, heat resistance, dimensional stability, and improved processing.

## Data Availability

Not applicable.

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
