# Peer review of "Engineering Polypropylene–Calcium Sulfate (Anhydrite II) Composites: The Key Role of Zinc Ionomers via Reactive Extrusion"

_polymers, 2023, doi:10.3390/polym15040799_

Round 1

Reviewer 1 Report

This is an excellent work in presenting the morphology and related mechanical fine-tuning of PP-based composites using modification with ZnDA. The methodology, results, underlying hypothesis and conclusions are presented in a exemplary way. The relevance is high both from academical and applications point of view as melt-mixing was used for the final composites. Publication is fully supported. I would like to offer two improvement suggestions to the authors:

Figure 4: Here only the structure of the 40% composites is presented. The 20% ones should also be presented. Perhaps by keeping only the higher magnification (right column) and using the space on the left for the other three samples?

The title needs to be adjusted. The generic way it stands ("recent advances"... "new fillers..."), implies that the manuscript is a review. It should be however modified to reflect the specific materials used herein. For example "Engineering polypropylene composites with calcium sulfate (anhydrite ii) modified with zinc diacrylate via reactive extrusion".

Minor points:
Line 15 & 704: performant --> performance
Line 20 & 122: customed --> custom

Reviewer 2 Report

In this manuscript, authors developed composites by filling PP with CaSO4 β-anhydrite II (AII) issued from natural gypsum. The effects of addition of up to 40 wt.% AII into PP matrix have been evaluated in terms of morphology, mechanical and thermal properties. They stated that, the PP–AII composites (without any modifier) as produced with internal mixers showed enhanced thermal stability and stiffness. At high filler loadings (40% AII), there was a significant decrease in tensile strength and impact resistance, therefore, tailor-made formulations with special reactive modifiers/compatibilizers (PP functionalized/grafted with maleic anhydride (PP-g-MA) and zinc diacrylate (ZnDA)) were developed. The manuscript presents some good data. I reviewed the manuscript in a critical manner and some of the comments are given below:

General comments

The manuscript is certainly a contribution of interest for “Polymers” and in principle within its specific scope; hence it is suitable for publication. The manuscript is sufficiently organized with clear novelty and well stated objectives. The quality of writing is more or less good with some grammar and spelling errors here and there. The English language usage should be checked by a fluent English speaker and/or a professional language editing service.

Moreover most of the results are consistent (e.g DSC , TGA , mechanical testing, etc) which indicated the absence of experimental errors during composites processing.

I recommend acceptance

Minor Specific comments

1. To back up the statement of tensile tests given in Fig. 7 authors need to provide FTIR spectra of all samples which are crucial for the understanding of the types of interfacial interaction between the filler and the PP.

2. In the same Fig.7 please include the stress-strain curves of the samples for better visualization

3. The elongation and break and toughness needs to be also reported

Other than that, overall; a good Ms. with well established work strategy.
